# Burnout in Nurses of an Intensive Care Unit during COVID-19: A Pilot Study in Portugal

**DOI:** 10.3390/healthcare11091233

**Published:** 2023-04-26

**Authors:** Cecília Almeida, Ana Filipa Poeira

**Affiliations:** 1Centro Hospitalar de Setúbal, 2900 Setúbal, Portugal; 2Instituto Politécnico de Setúbal, ESS, NURSE’IN-UIESI, 2900 Setúbal, Portugal

**Keywords:** burnout, critical care, COVID-19, intensive care unit, mental health, nurses

## Abstract

Background: This study aimed to evaluate burnout in intensive care unit nurses and describe the relationship between burnout dimensions and sociodemographic and professional variables. Methods: A pilot study was carried out with 29 intensive care nurses during the COVID-19 pandemic. The data were collected using an online questionnaire consisting of a sociodemographic characterization and the Maslach Burnout Inventory Human Services Survey. In the comparative statistical analysis, the nonparametric Mann–Whitney and Kruskal–Wallis tests were used (*p* < 0.05). Results: The Emotional Exhaustion dimension represented an average of 27.9 ± 10.4, and the Personal Fulfillment of 30.8 ± 7.0 was considered high. Regarding Depersonalization, medium-level results (8.1 ± 6.0) were found. There was a prevalence of 41.3% of high levels of burnout. There was no statistically significant difference in the comparison between the three dimensions and sociodemographic and professional variables (*p* > 0.05). Conclusions: The global results point to high levels of burnout in nurses, with greater expression in the Emotional Exhaustion and Professional Fulfillment domains. There were no differences between groups, demonstrating homogeneity in the team’s level of burnout. The incidence of burnout in this study was higher than that identified in other studies carried out in Portugal with health professionals. This pilot study will allow the defining of strategies for reducing burnout and, consequently, a proposed extension of the study to other organization wards.

## 1. Introduction

Work is often of central importance to a person’s identity and self-worth, so it is not a surprise that it has the potential to improve or harm a person´s well-being. Worker well-being becomes a concern of organizations and “comes down to the general well-being of working people” [1]. The World Health Organization included burnout in the 11th Revision of the International Classification of Diseases (ICD-11) as an occupational phenomenon, defining it as a syndrome resulting from chronic stress in the workplace that has not been successfully managed [2]. Burnout is characterized by three dimensions: Emotional Exhaustion, Depersonalization, and Personal Fulfillment. Emotional Exhaustion is understood as a depletion of the person´s emotional, moral, and psychological resources and is considered the central element of burnout. Depersonalization represents an affective distance or emotional indifference towards others, namely those who are the reason for the professional activity (patients, clients, persons, family, community). Finally, Personal Fulfillment expresses a decrease in feelings of competence and pleasure associated with the performance of a professional activity [3] (p. 153). This syndrome is prevalent in healthcare environments [4], and nurses, as the largest group of healthcare professionals, are at risk, which carries profound negative implications for nurses, patients, colleagues, and healthcare organizations [5]. The literature is vast on the impact on health workers’ job satisfaction, burnout, and turnover intention. There has been a consensus that there are losses regarding the quality of care provided [6]. The stress level and physical and mental fatigue vary depending on the ward in which the nurse performs, and the emergency department, the intensive care unit (ICU), the operative room, and post-anesthesia potentiate this risk [7,8,9].

In the case of the ICU, nurses provide the highest level of care to critically ill patients. Therefore, they must expend much energy to provide care and meet critically ill patients’ needs [10]. Although many nurses are, therefore, exposed to situations of extreme emotional anguish (premature death, patient suffering, poor prognosis), not all of them can manage the emotional stress resulting from this exposure, which is why support is crucial to prevent them from developing inappropriate and harmful responses, and that may subsequently evolve into burnout [10,11]. Furthermore, the emergence of new working orders due to the pandemic has made working environments extremely stressful [12]. As expected, an increase in exposure to risk factors enhances the development of burnout syndrome, making it increasingly necessary to address this issue [13].

Suffering at work begins when there is a failure in the intermediation between nurses’ expectations and the reality imposed by the work organization, since the nurses establish, in most cases, an affinity and a greater bond during the provision of care because the treatment of people is generally prolonged [14]. In this sense, they need to adopt coping strategies to face and deal adequately with the stressors mentioned above and, consequently, avoid psycho-emotional disturbances. Health professionals had considerable anxiety, insomnia, stress, and burnout due to the COVID-19 pandemic, and nurses had a higher rate of affective symptoms [15]. In addition, the COVID-19 pandemic has pushed an already overstretched nursing workforce into largely uncharted territory [16]. Furthermore, in Portugal, the shortage of nurses is a reality and concern, and hospitals and other health organizations continue to have difficulty recruiting [17]. In the Organisation for Economic Co-operation and Development report, on the country’s health profile, it was clear that one of the challenges of the National Health Service is to retain its professionals: “In fact, recent years have seen a wave of emigration among health care workers, particularly nurses. The future challenge for the National Health Service is to be able to maintain the motivation of its workforce, and to contain and reverse the drain of professionals. The low number of nurses, however, is not likely to grow in the near future: while the number of medical graduates has increased consistently over time, the number of nurse graduates has been decreasing since 2009” [18] (p. 14). With the pandemic, this reality has worsened.

Given this, implementing preventive measures to reduce the impact of risk factors and the development of burnout through raising awareness, education on the problem, and assessment of the team and specific intervention to reduce burnout become essential [13].

From this perspective, this study aimed to assess the burnout level in nurses in an ICU and describe the relationship between burnout and sex, age, category, and professional experience. Both organizations from a macro perspective and nurse managers from a micro perspective should implement management projects that identify and manage the factors that induce burnout, that implement interventions for the reduction and prevention of burnout, and that equally assess the impact of this project on the quality of life and satisfaction of its professional team. Thus, this pilot study intended to identify the influence of the pandemic on the burnout level of nurses.

## 2. Materials and Methods

A pilot study was carried out in an ICU with a simple random sample of 29 nurses. The study was defined by the inclusion criteria of being an ICU nurse during COVID-19 pandemic. All nurses were invited to participate, in the case of a non-probabilistic sample. The total of nurses working in this ICU was 47, corresponding to a response rate of 61.7%. As a data collection instrument, a questionnaire was used. The first part consisted of questions related to sociodemographic and professional data, and the second part aimed at assessing levels of burnout, consisting of the Maslach Burnout Inventory Human Services Survey (MBI-hss) [3,19].

To verify the validity of the MBI-hss, a factor analysis was performed, including its 22 items with Varimax orthogonal rotation. The Kaiser–Meyer–Olkin test (KMO = 0.60) and Bartlett´s Sphericity test (value of 431.12; *p* = 0.00) allowed verifying that the variables were significantly correlated and the adequacy of the scale for the study. The scale had an average internal consistency with a Cronbach’s alpha of 0.74 for all 22 items.

The MBI-hss is an inventory of 22 items that report work-related feelings, divided into three scales: Emotional Exhaustion, Depersonalization, and Personal Fulfillment. The answer is given based on the frequency with which each feeling occurs on a seven-position Likert-type scale between “Never” (0) and “Every day” (6). For each dimension, the following scale items were considered: Emotional Exhaustion (Items 1, 2, 3, 6, 8, 13, 14, 16, 20); Depersonalization (Items 5, 10, 11, 15, 22); Personal Fulfillment (Items 4, 7, 9, 12, 17, 18, 19, 21) [3,19].

The internal consistency was also evaluated for each of these dimensions, presenting satisfactory internal consistency for all of them. Cronbach´s alpha coefficients based on the sample were α (Emotional Exhaustion) = 0.89; α (Depersonalization) = 0.75; α (Personal Fulfillment) = 0.78.

The global burnout score was then converted into classes of burnout following the recommendation of Maslach et al.: no burnout/reduced burnout for mean scores less than two; moderate burnout for average scores across (two- three); high burnout for higher average scores or equal to three [20].

The sociodemographic questionnaire and the MBI-hss were completed online using Google Forms, available by email. The questionnaires were applied from November 2021 to the end of January 2022.

The statistical analysis of the data was performed through a descriptive analysis and a multivariate analysis using the nonparametric Mann–Whitney and Kruskal–Wallis tests (*p* < 0.05).

### Ethical Considerations

Informed consent was obtained by asking all participants to select “yes” or “no” at the beginning of the online survey to consent to participate. To use the MBI-hss, authorization was required from the authors responsible for translating and validating the instrument for the Portuguese population [21]. The Hospital’s Ethics Committee approved the study (No. 37/2021).

## 3. Results

The study used a random sample of 29 nurses from a total of 47 nurses working in the ICU. In the sociodemographic and professional characterization of the sample, it was found that most of the sample belonged to females (*n* = 23), representing 79.3%, and the remaining 20.7% belonged to males (*n* = 6).

Regarding age ranges, it was found that 41% of individuals were between 30 and 39 years old. It should be noted, however, the existence of a balance in the age distribution in the sample, with 31% (*n* = 9) between 20 and 29 years old and 28% (*n* = 8) between 40 and 49 years old.

Most of the sample belonged to the category of Nurse (*n* = 20), equivalent to 69%, with the category of Specialist Nurse represented by 31% of the same (*n* = 9).

Regarding the length of service, it was found that most of the sample had less than five years of professional experience, with 37.9% (*n* = 11), but it should be noted that 34.5% (*n* = 10) had between 11 and 20 years and 20.7% (*n* = 6) between 5 and 10 years of service.

In the analysis of the MBI-hss questionnaires, the totals of each dimension (Emotional Exhaustion, Depersonalization, and Personal Fulfillment) are accounted for in Table 1, and the overall results were subsequently verified.

Regarding Emotional Exhaustion, it was found that most nurses (*n* = 14) had a high level, with 48.3% of individuals. The remaining sample was distributed with *n* = 9 (31.0%) at the medium level and only *n* = 6 (20.7%) at the low level.

In the distribution of the Depersonalization subscale, it was verified that 37.7% of the individuals (*n* = 11) had a low level in this dimension and the same number a high level.

The Personal Fulfillment dimension showed the highest levels of burnout with 65.5% of the sample (*n* = 19), which points to low Personal Fulfillment. It should be noted that only 6.9% (*n* = 2) had a low level in this dimension.

In the overall evaluation of the results, a total average of 66.8 ± 13.2 was obtained, which means an average score of 3.1 ± 0.6, which indicated a high burnout score in the sample [16]. Most of the nurses (*n* = 16, 55.2%) had medium levels of burnout. However, a prevalence of 41.4% (*n* = 12) of individuals with high levels of burnout was verified.

In the multivariate analysis, considering the sample size (*n* < 30), nonparametric tests were performed for comparative effects to identify differences. The levels of burnout in the three dimensions were not significantly different between women and men (*p* > 0.05), as verified by the interpretation of the *p*-value presented in Table 2. It was also verified that the relationship between category (Nurse or Specialist Nurse) and the levels of burnout in the three dimensions was not significantly different by interpreting the *p*-value (Table 2).

Comparing age and the dimensions of Burnout, the Kruskal–Wallis test showed that there are no statistically significant differences between groups. However, it was found that nurses aged between 40 and 49 years old presented a higher value in terms of Emotional Exhaustion. In turn, in the Depersonalization dimension, nurses aged 30 to 39 years presented a higher value. This group of nurses also had the lowest value in the Personal Fulfillment dimension (Table 3). Regarding professional experience and dimensions of burnout, it is also verified that there were no statistically significant differences between the groups (Table 3).

## 4. Discussion

The literature defines nurses as a highly vulnerable population for burnout, either because of the inherent care they provide or because of the increasing work demands they are subject to and the workplace [10]. In a multicenter comparative study on burnout among nurses, it was found that the workplace plays an essential role in the development of this syndrome, with emphasis on the differences between the workday of the emergency or intensive care nurse and that of the primary care nurse [22]. The current study presented data on the levels identified in the nursing team of an intensive care unit in the context of the COVID-19 pandemic. This reality brought physical and psychological conditions that triggered higher stress levels and can aggravate the risk [4,11]. The impact of high levels of stress can result in increased burnout and shortages of nurses, which can make the public find it difficult to access health services. The COVID-19 pandemic has, thus, highlighted critical care nurses’ incredible value and importance. They are highly trained professionals with specialized knowledge and improved skills with each patient served. However, such qualities are not replaceable; not everyone can play the role of a critical care nurse [23]. Because of this, it becomes essential to retain these nurses, minimize the costs, and ensure the quality of nursing care [24]. The nurse practice environment is correlated with intent to leave, especially in management and staffing, and resource adequacy dimensions, which indicates that providing sufficient resources and support at work could decrease the turnover intention of nurses [25].

A multicenter study carried out during the COVID-19 pandemic showed “that nurses’ workload in intensive care had increased significantly related to complex procedures, such as invasive ventilation of patients and the use of Extracorporeal Membrane Oxygenation (ECMO), the pathophysiology of the disease and the occurrence of adverse events” and “all these factors, inherent to work in intensive care, reveal the profound changes and adaptations that have taken place in the services in the face of the pandemic, becoming stressors and potentially aggravating health” [26] (p. 8).

The results of the study pointed to moderate to high levels in nurses working in this ICU, respectively 55.2% and 41.4%. These are superior to the results obtained in another study carried out with nurses during the pandemic, with moderate levels of 40.9% and high levels of 38.5% [27]. Moreover, consistent with the results of a search about factors related to nurses’ burnout during the first wave of COVID-19, where levels of burnout and exhaustion were almost 90%, the participants met the criteria for medium/high burnout [28].

The results obtained in the study by Oliveira, although lower than the explained study, still demonstrated an increase in burnout levels compared to studies carried out with health professionals before the COVID-19 pandemic [27]. In a systematic review and meta-analysis, it was concluded that nurses in a high-risk clinical environment, such as a COVID-19 unit, associated with inadequate, defective material and human resources and low safety in caring for patients, had higher levels of burnout, demonstrating that occupational factors primarily affecting these professionals during the pandemic [4].

Although several predictors’ factors are recognized in the National Health System for burnout, such as, for example, the lack of investment and dissatisfaction with working conditions, effectively, the public health emergency put additional pressure on nurses and may have contributed to this increase in burnout levels [5,27].

Regarding the analysis of each of the burnout dimensions, it appears that Emotional Exhaustion and the decrease in Personal Fulfilment were the ones with the highest scores, which was also verified in the study by Borges et al. [22]. It was found in a study carried out with nurses working in Portugal that their sleep quality and symptoms of depression, anxiety, and stress showed a positive variation concerning the COVID-19 outbreak [29]. It should be noted that no correlation was found between sociodemographic and professional characterization variables with the three dimensions of burnout demonstrating the homogeneity of the levels evaluated in the team. This fact was not verified in a systematic review and meta-analysis in which sociodemographic factors, such as sex, age, and education, influence nurse burnout [5]. In another study, the analysis also showed that no variable tested was a predictive factor for Emotional Exhaustion. However, age and professional experience were considered significant predictive factors for Personal Fulfillment [22]. On the other hand, Montes-Berges and Fernández-García found that having worked in the ICU for more than five years, caring for COVID-19 patients for more than six months, and witnessing the deaths of patients with this disease during the journey of work were aggravating factors of nursing team burnout [30].

Concerning Depersonalization in ICU nurses, this may be the last dimension manifested, since it is more difficult to identify Depersonalization in an environment where there is less space to demonstrate empathy with another; a relationship between the nursing team and the patient may be absent due to the severity of the clinical status (coma/unconsciousness). Thus, it may lead to greater difficulty in attributing indifference to patients, which could mask the presence of Depersonalization [31]. However, in this study, it was found that 37.7% of individuals had a high level in this dimension, which may also indicate the severity of the problem.

Increased burnout and worker shortages will negatively affect the healthcare system and the ability of organizations to deliver the best healthcare, which should aim to improve the patient experience, improve the population´s health, and reduce costs. For this, the well-being and prevention of burnout of care teams must be considered a fundamental prerequisite [32]

Another study demonstrated that burnout fully mediated the relationship between organizational support and intent to leave. Thus, when ICU nurses perceive they have organizational support, they will be less likely to suffer from burnout symptoms leading to turnover intentions. These findings demonstrated the crucial ramifications of workplace support [33]. Emotional exhaustion is strongly correlated with nurses’ intent to leave, and burnout is an especially critical factor, inducing the sense to leave [22].

Although psychological risk cannot be completely avoided during pandemics due to its emergent nature and the initial challenges faced in implementing strategies can never be completely eliminated [34], the literature considers that the knowledge of both risk and protective factors is a valuable resource and can be adopted not only to cope with the still-active COVID-19 pandemic, but also future potential other periods that may increase the demand for ICU care [35]. Leaders and organizations must perceive the signals for awareness and timely intervention [36]. Raising awareness and educating non-psychiatric medical teams for mental health assessment can be crucial to enable timely diagnosis [29].

An organizational change is necessary. Organizational solutions to burnout in nurses may include initiatives focused on redesigning the workflow to reduce administrative burden or strategies to improve interprofessional teamwork [37]. Organizational support can be seen as a strategy to reduce the negative outcomes of pandemics such as COVID-19. This can take many different forms: they can be more structural (offering flexible hours), while others can be more supportive (showing gratitude for your efforts) [33].

The literature indicates that nurse leaders play a crucial role in supporting the mental health needs of the human resources they manage, minimizing burnout and ensuring the sustainability of the nursing workforce during the COVID-19 pandemic [38]. In view of this, nursing managers need to pay much more attention to the health professional and design effective and systematic long-term programs to identify the sources of stress and resolve them to manage not only the consequences of the recent pandemic outbreak, but also to be able to control possible future pandemic situations [39]. Awareness of the problem allows the implementation of early and appropriate support measures, such as normalizing emotions, clear communication, and meeting basic needs. In addition, the investment and implementation of mental well-being strategies and psychological interventions should be prioritized and encouraged to improve the health of nurses [40].

Given the results obtained, it is possible to go from assuming that the nurses were exhausted to affirming that this is a reality. In view of these results, we propose the implementation of a project based on the intervention of the nurse manager of the ICU in question through standardized assessment tools, prevention strategies, and the minimization of the consequences of burnout syndrome in this team. This pilot study highlights the need for change and promotes it.

In this context, a global assessment of burnout levels for all nurses and for other health professionals at the institution is still desirable, aiming to diagnose the existing reality and subsequently encourage the development of organizational strategies that promote the occupational health and well-being of these professionals. Organizations should also identify the characteristics and resilience of nurses to direct interventions that allow nurses to build their resilience [41,42].

## 5. Limitations of the Study

This pilot study has significant limitations that make the results preliminary and only suggestive. The study’s major limitation is the sample size and the need to provide deep insight or contribute to the discussions about burnout. This fact affects representativeness, but due to the high rate of burnout that was found, it warns about a possible problem and the possible need to replicate the study in the remaining services of the hospital organization. Therefore, it is essential to consider extending the research in the area to the entire organization for a better diagnosis of the situation regarding the levels of burnout in nurses during the COVID-19 pandemic. Another limitation is that, as the collection of data was individualized to only one ward, it could have been complemented with the collection of qualitative data. A mixed study would have allowed a better understanding of the problem from an inductive perspective. The cross-sectional design was also assumed to be a limitation since it does not allow for the monitoring of individuals to verify the impacts of the pandemic on mental health, and it would be relevant to carry out a longitudinal study that would allow analyzing the evolution of the mental health status of nurses, as well as the evolution of the impact of each factor associated with mental health outcomes over time.

## 6. Practical Implications

Organizational intervention projects must be implemented considering the specific characteristics of the services and the particularities of the teams. In this team, specifically, interventions aimed at the domain of Emotional Exhaustion and Personal Fulfillment will be the priority.The coping strategies mobilized by health professionals can be both organizational and personal, so achieving more individualized and team-centered results will allow an awareness of the need to contribute individually to construct flexible and healthy work environments.Further studies to assess the mental health of this professional group and monitor the long-term effects of coping with COVID-19 to understand its real impact are important.

## 7. Conclusions

The study’s overall results point to high burnout levels in nurses, with greater expression in the Emotional Exhaustion and Personal Fulfillment domains. It was concluded that, for all variables for sociodemographic and professional characterization, there were no statistically significant differences between groups in the three dimensions of burnout, which ended up reinforcing what the descriptive analysis demonstrated: that there is a homogeneity in the group that is characterized by the whole team with high levels of burnout, regardless of their characteristics and particularities. The COVID-19 pandemic, which forced a profound restructuring and a new resource dynamics, confronted professionals with enormous pressure and demand. The pandemic has shown that the health sector represents a growing workforce that needs continuous investment due to the priority given to healthcare and qualified human resources. In this regard, burnout in this professional group is a problem that requires a comprehensive approach based on organizational strategies.

## Figures and Tables

**Table 1 healthcare-11-01233-t001:** Distribution by dimension Emotional Exhaustion, Depersonalization, and Personal Fulfillment.

Burnout Level	Emotional Exhaustion *n* (%)	Depersonalization *n* (%)	Personal Fulfillment *n* (%)
Low	6 (20.7%)	11 (37.9%)	2 (6.9%)
Medium	9 (31.0%)	7 (24.1%)	8 (27.6%)
High	14 (48.3%)	11(37.9%)	19 (37.9%)

**Table 2 healthcare-11-01233-t002:** Mann–Whitney test between dimensions of burnout and sex and category.

Variables		Mann–Whitney Test		
		Emotional Exhaustion	Depersonalization	Personal Fulfillment
Sex	Man (*n* = 6)	69.5	98.5	104.0
	Women (*n* = 23)	365.5	336.5	331.0
	Mann–Whitney U	48.500	60.500	55.000
	*p*-value	0.269	0.646	0.449
Category	Nurse (*n* = 20)	13.43	15.38	15.63
	Specialist Nurse (*n* = 9)	18.5	14.17	13.61
	Mann–Whitney U	58.500	82.500	77.000
	*p*-value	0.137	0.723	0.554

*p* > 0.05 (not significant).

**Table 3 healthcare-11-01233-t003:** Kruskal–Wallis test between dimensions of burnout and age and professional experience.

Variables		Kruskal–Wallis Test		
		Emotional Exhaustion	Depersonalization	Personal Fulfillment
Age	20–29 (*n* = 9)	12.56	12.22	17.44
	30–39 (*n* = 12)	15.88	18.33	11.79
	40–49 (*n* = 8)	16.44	13.13	17.06
	H	1.099	3.203	2.943
	*p*-value	0.577	0.202	0.230
Professional Experience	<5 years (*n* = 11)	11.32	13.64	18.00
	5–10 years (*n* = 6)	18.25	22.58	13.92
	11–20 years (*n* = 10)	17.15	12.50	10.50
	21 or more years (*n* = 2)	14.75	12.50	24.25
	H	3.580	6.145	6.680
	*p*-value	0.311	0.105	0.083

*p* > 0.05 (not significant).

## Data Availability

Data are unavailable due to ethical restrictions.

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
