# Peer review of "Burnout in Nurses of an Intensive Care Unit during COVID-19: A Pilot Study in Portugal"

_healthcare, 2023, doi:10.3390/healthcare11091233_

Round 1
Reviewer 1 Report
Dear Authors:
Thank you for allowing me to review this manuscript. It is an interesting, logical, well-understood manuscript. which, however, needs improvement. There is no detailed description of the study group and inclusion criteria (which nurses were invited to participate in the study?). The most important limitation of this study is too small a study group - 29 nurses. I suggest changing the topic of the work to a pilot study or expanding the study to a larger group. Statistical methods are important and correctly applied. I propose to break down the conclusions in points and add practical implications.
Author Response
Dear Reviewer,
Thank you for your time, work, and contributions that will contribute to the publication of the high-quality paper.
Thus, we sent a table of discrimination with the changes made in the manuscript.
Best regards,
Cecília Almeida e Ana Poeira

Reviewer 2 Report
While the study seems sound in terms of objectives and research methods, the very small sample size does not provide deep insight or contribution to the discussions about burnout. The explanatory variables included do not provide theoretical depth on the story of the paper. The discussion is too broad and non- specific.
Author Response

(The authors gave the same response as above.)

Reviewer 3 Report
As you mentioned the sample size is a limitation of the study. Starting from a participation of the 90% of the personnel of the Unit / Department it would have been sufficient to give to the readers an idea of the burnout level of the ICU of the hospital. But having only the 60% sample of the staff I don't think could give a view of the burnout situation. But it is your choice to go on with this work to the publication or not.
Author Response

(The authors gave the same response as above.)

Reviewer 4 Report
The authors presented interesting results of study covered burnout in nurses of an intensive care unit during COVID-19. I have some comments:
1) The study group is quite small, which can affect the results. It is necessary to discuss this fact as a limitation.
2) How do we ideologically compare the results in Table 2 and 3? Regression analysis, or another version of multivariate analysis, is preferable if there is enough statistical power.
Author Response

(The authors gave the same response as above.)

Round 2
Reviewer 1 Report
Thank you for making corrections to the manuscript.
Author Response
Dear Reviewer,
Thank you for your comment.
Best regards,
Cecília Almeida e Ana Poeira
Reviewer 4 Report
I see an improvement in the quality of the article. However, I do have comments:
1) For the result from Table 3, it is advisable to perform a post-hoc analysis.
2) Why don't the authors want to do a multivariate analysis by combining the variables of Tables 2 and 3? I don't understand how Age, Sex, Category and Professional Experience are related. Which of these factors are more significant for burnout?
Author Response
Dear Reviewer,
Thank you for your comments.
We hope we can justify our methodological options regarding the statistical treatment of the data, given that these options were intended to maintain the honesty and veracity of what was done.
We have attached a file with the answer.
Best regards,
Cecília Almeida e Ana Poeira
